# The Flow Cytometric Evaluation of B- and T-Lymphoblastic Leukemia/Lymphoma

**DOI:** 10.3390/cancers17071111

**Published:** 2025-03-26

**Authors:** David M. Dorfman

**Affiliations:** Department of Pathology, Brigham and Women’s Hospital, Harvard Medical School, 75 Francis Street, Boston, MA 02115, USA; ddorfman@bwh.harvard.edu; Tel.: +1-617-732-7518

**Keywords:** immunophenotypic analysis, hematogones, thymus, thymoma, NK-lymphoblastic leukemia/lymphoma, early T-precursor lymphoblastic leukemia/lymphoma (ETP-ALL), acute leukemia of ambiguous lineage, mixed-phenotype acute leukemia, acute undifferentiated leukemia

## Abstract

This paper describes how flow cytometry may be used to aid in the diagnosis and subtyping of B- and T-lymphoblastic leukemia/lymphoma. It presents the typical flow cytometry findings in these diseases and their various subtypes, as well as in non-neoplastic conditions and other diseases that need to be distinguished from B- and T-lymphoblastic leukemia/lymphoma.

## 1. Introduction

Lymphoblastic leukemia/lymphoma is a neoplasm of precursor B or T lineage lymphoid cells that usually involves the bone marrow and peripheral blood, and may involve nodal and/or extranodal sites. It occurs primarily in children, who typically present with bone marrow failure and/or a high white blood cell count. When the disease is primarily extramedullary, it is referred to as lymphoblastic lymphoma. Lymphoblastic lymphoma is more common in T-lymphoblastic leukemia/lymphoma than in B-lymphoblastic leukemia/lymphoma, typically presenting as a mediastinal mass. In contrast to acute myeloid leukemia (AML), there is no minimum blast percentage in the blood or bone marrow required for a diagnosis of lymphoblastic leukemia. B-lymphoblastic leukemia accounts for approximately 85% of pediatric and 75% of adult lymphoblastic leukemias, and T-lymphoblastic leukemia accounts for the remainder, although it accounts for 85–90% of lymphoblastic lymphomas [1,2,3].

The diagnostic approach for B- and T-lymphoblastic leukemia/lymphoma is multifactorial, and includes morphologic assessment, immunophenotypic analysis, usually by flow cytometry, and genetic analysis, including cytogenetics and FISH analysis, as well as molecular diagnostic analysis. This review will focus on flow cytometric immunophenotypic findings in B- and T-lymphoblastic leukemia/lymphoma.

## 2. B-Lymphoblastic Leukemia/Lymphoma

B-lymphoblastic leukemia/lymphoma is characteristically positive for B cell markers, including CD19, CD79a, and CD22, which can all be assessed by flow cytometric analysis. The neoplastic cells typically exhibit an immature B cell immunophenotype, which includes the expression of markers of B cell immaturity, such as terminal deoxynucleotidyl transferase (TdT), stem/progenitor cell marker CD34, low-level expression of CD45, low to absent expression of CD20, a marker of mature B cells, and the absence of surface immunoglobulin kappa or lambda light chain expression [2,4]. Cases of B-lymphoblastic leukemia/lymphoma may express markers reflective of the various stages of B cell maturation, including the earliest, pro-B stage (CD19, cytoplasmic CD79a, cytoplasmic CD22, TdT, negative for CD10, and IgM), the intermediate or common-B stage (CD10 positive and negative for IgM), and the later pre-B stage (positive for cytoplasmic and surface IgM) [2,4]. It is important to note that cases of acute myeloid leukemia may express B cell markers, including CD19, cytoplasmic CD79a, and PAX5, particularly in acute myeloid leukemia with t(8;21)(q22;q22.1) [5], and cases of T-lymphoblastic leukemia/lymphoma may express CD10 and low-level CD79a (see below) [6]. Flow cytometric findings from a typical case of B-lymphoblastic leukemia/lymphoma, from a 59-year-old woman with leukocytosis and 91% circulating blasts, are shown in Figure 1. In that case, the neoplastic cells were dimly positive for CD45, CD34, and CD20, positive for HLA-DR, CD38, co-expressed B cell markers CD19 and CD10, and were negative for myeloid antigens CD15 and CD33. The neoplastic cells were negative for additional myeloid and monocytic markers (CD13, CD11b, CD14, and CD117) and T cells markers (CD3, CD5, and CD7) [4].

B-lymphoblastic leukemia/lymphoma may express one or more markers of myeloid lineage, including CD13 and CD33, but is usually negative for moderate to high levels of myeloperoxidase expression (which can be assessed by flow cytometric analysis, cytochemical, or immunohistochemical staining) [2,7]. High-level expression of myeloperoxidase in acute leukemia expressing B cell markers favors a diagnosis of B/myeloid mixed-phenotype acute leukemia (MPAL) or acute myeloid leukemia, which are discussed elsewhere in this publication [8]. In one study, 86.5% of cases of B-lymphoblastic leukemia (88.5% of pediatric cases and 82% of adult cases) expressed one or more myeloid antigens, which was typically partial or dim [7]. The most frequently expressed myeloid antigen was CD13, present in 54.5% of cases, followed in decreasing frequency by CD33, present in 43.0% of cases, CD15, present in 36.0% of cases, and CD11b, present in 20.0% of cases [7]. Six additional myeloid antigens (CD36, CD64, CD14, CD16, CD117, and myeloperoxidase) were altogether present in 17.5% of cases. The majority of cases expressed one or two myeloid antigens, present in 28.5% of cases and 39.0% of cases, respectively, 13.0% of cases expressed three myeloid antigens, 4.5% of cases expressed four myeloid antigens, and 1.5% of cases expressed five myeloid antigens [7]. The expression of myeloid antigens CD13 and CD33 in B-lymphoblastic leukemia/lymphoma is typically observed in B-lymphoblastic leukemia/lymphoma with BCR::ABL1 fusion [2]. The expression of T cell antigens, including CD2, CD4, CD5, CD7, and monocytic antigens, has also been reported in B-lymphoblastic leukemia/lymphoma. In one study, 10.0% of cases of B-lymphoblastic leukemia/lymphoma expressed one or more T cell antigens: CD4 in 5.0% of cases, CD5 in 2.2% of cases, CD7 in 2.2% of cases, and CD2 in 0.6% of cases [7].

B-lymphoblastic leukemia/lymphoma has been reported to be negative for CD45 in as many as 12.9% of cases in children, in contrast to 3.7% of cases of T-lymphoblastic leukemia/lymphoma [2,9,10]. The highest frequency of CD45-negative B-lymphoblastic leukemia/lymphoma cases was found in cases with an intermediate, common-B stage immunophenotype (15.1%), followed in frequency by cases with a later pre-B stage immunophenotype (7.8%), and the earliest, pro-B stage immunophenotype (7.2%) [9]. Because flow cytometric identification of the neoplastic cells typically includes gating on and the study of CD45 positive cells, it is possible that cases of CD45-negative B-lymphoblastic leukemia/lymphoma may be missed [10]. Complete examination of ungated dot plots is essential in suspected cases of B-lymphoblastic leukemia/lymphoma in which neoplastic cells have not been identified by typical flow cytometric analysis [10]. Figure 2 shows a case of CD45-negative B-lymphoblastic leukemia/lymphoma from a 4-year-old girl with pancytopenia. Initial flow cytometric analysis performed on CD45 positive cells, shown in green, revealed a mixed population of CD3 positive T cells, with subsets positive for CD4, CD8, and CD19 positive B cells, with subsets positive for polyclonal surface immunoglobulin kappa and lambda light chain expression. Subsequent flow cytometric analysis performed on CD45-negative cells, shown in pink, revealed a population of cells positive for CD19, CD10, and CD20 (dim), and negative for surface immunoglobulin kappa and lambda light chains, consistent with B lymphoblasts and involvement by B-lymphoblastic leukemia/lymphoma [10].

The fifth edition of the World Health Organization classification of hematolymphoid tumors recognizes multiple subtypes of B-lymphoblastic leukemia/lymphoma with specific genetic abnormalities, which may correlate with specific immunophenotypic findings as well as prognosis [2]. A number of these subtypes may be suggested by flow cytometric immunophenotypic findings; however, complete genetic characterization is required for definitive diagnosis (Table 1). As noted above, B-lymphoblastic leukemia/lymphoma with BCR::ABL1 fusion often exhibits aberrant expression of myeloid antigens CD13 and CD33, although these findings are not specific for this subtype (see below). An example of a case of B-lymphoblastic leukemia/lymphoma with BCR::ABL1 fusion is shown in Figure 3. The neoplastic cells are dimly positive for CD45, positive for CD34, and positive for B cell markers CD19 and CD10 show dim-to-negative staining for CD20, and coexpress myeloid markers CD33 and CD13. B-lymphoblastic leukemia/lymphoma with BCR::ABL1 fusion has also been reported to exhibit a decreased expression of CD9 and CD81 when compared with cases of B-lymphoblastic leukemia/lymphoma without cytogenetic alterations or with other cytogenetic abnormalities [11]. B-lymphoblastic leukemia/lymphoma with *BCR::ABL1*-like features exhibits a *CRLF2* gene rearrangement in 50% of cases, leading to an overexpression of CRLF2 protein, which can be assessed by flow cytometric analysis. B-lymphoblastic leukemia/lymphoma with *KMT2A* rearrangement typically has a CD19-positive, CD10-negative, CD24-negative, and TdT-negative immunophenotype, as well as positivity for myeloid markers CD15 and CD65. An example of a case of B-lymphoblastic leukemia/lymphoma with *KMT2A* rearrangement is shown in Figure 4. The neoplastic cells are dimly positive for CD45, positive for B cell marker CD19, negative for B cell markers CD10 and CD20, and coexpress myeloid marker CD15 (dim). B-lymphoblastic leukemia/lymphoma with *ETV6::RUNX1* fusion is often negative for CD9, CD20, and CD66c, and positive for myeloid markers CD13 and CD33. B-lymphoblastic leukemia/lymphoma with *TCF3::PBX1* fusion is typically strongly positive for CD9, dim-to-negative for CD34, and dim-to-negative for CD20. B-lymphoblastic leukemia/lymphoma with *TCF3::PBX1* fusion has also been reported to overexpress CD58 and CD81 when compared with cases of B-lymphoblastic leukemia/lymphoma without cytogenetic alterations or with other cytogenetic abnormalities. B-lymphoblastic leukemia/lymphoma with *IGH::IL3* fusion may show expressions of myeloid markers CD13 and/or CD33. B-lymphoblastic leukemia/lymphoma with DUX4 rearrangement typically is positive for CD371 and CD2. Cases may exhibit expressions of monocytic marker CD14, gains of CD45 and CD33, and losses of B cell antigens after induction chemotherapy or at the time of diagnosis [2].

Normal B cell precursors (hematogones) need to be distinguished from residual B-lymphoblastic leukemia/lymphoma following chemotherapy. Normal B cells at the pro-B cell stage (Type I hematogones) are positive for early B cell markers CD19, CD10, CD34, CD38, and TdT, and have low-level expression of CD45 and lack expression of CD20. As normal B cells mature to Type II hematogones, they remain positive for CD38, the expression of CD10, CD34, and TdT decreases, and the expression of CD20 and CD45 increases. With further maturation to Type III hematogones, there is loss of expression of CD10, decreased expression of CD38, and increased expression of CD20 and CD45. These cells may also exhibit low-level expression of CD5 [4,12]. Distinguishing minimal residual B-lymphoblastic leukemia/lymphoma from hematogones relies on the observation in residual leukemic blasts of discrete rather than variable expression of B cell markers that vary in expression during normal B cells development, such as CD10, CD38, CD20, and CD34, as well as the expression in leukemic blasts of any aberrant markers, such as myeloid or T cells markers, noted at the time of diagnosis (Table 2). Complete immunophenotypic characterization of B-lymphoblastic leukemia/lymphoma at the time of diagnosis, including the identification of any aberrant antigen expression, is essential for accurate subsequent minimal or measurable residual disease (MRD) detection by flow cytometric analysis. The flow cytometric findings from a typical case of B-lymphoblastic leukemia/lymphoma, shown in Figure 1, include uniform positive expression of CD10 and CD38, in contrast to the variable expression of these markers typically seen in hematogones. Additional flow cytometric markers that may be helpful for B-lymphoblastic leukemia/lymphoma MRD assessment include CD304, CD73, and CD86, which are expressed by leukemic blasts in a significant number of cases of B-lymphoblastic leukemia/lymphoma, and can be used to distinguish them from hematogones and normal B cells [13,14,15]. Additionally, the leukemic blasts in B-lymphoblastic leukemia/lymphoma concordantly express CD123 and CD34 in the majority of cases and are double negative for these markers in a minority of cases, which can be used to distinguish them from hematogones, which typically display discordant expressions of these two markers: early hematogones are typically CD34+/CD123− and late hematogones are typically CD34−/CD123+ [16,17].

In B-lymphoblastic leukemia/lymphoma patients who have received immunotherapeutic agents that target CD19, such as Blinatumomab and anti-CD19 chimeric antigen receptor cells (CAR-Ts), the blasts may exhibit decreased or absent CD19 at the time of disease relapse [18,19]. Anti-CD22 immunotherapeutic agent Inotuzumab may lead to decreased or absent CD22 expression in B-lymphoblastic leukemia/lymphoma blasts at the time of relapse. In patients treated with these agents, it is important to recognize that normal and neoplastic B cells may be negative for CD19 and/or CD22. In patients treated with agents targeting CD19, alternative B cell markers, such as CD22 and CD24, can serve to identify normal and neoplastic B cells. CD24 is also expressed by neutrophils, so an approach to exclude neutrophils from analysis, such as excluding CD66b positive cells, may be employed. In patients treated with an agent targeting CD22, CD24 may serve to identify normal and neoplastic B cells; however, not all cases of B-lymphoblastic leukemia/lymphoma are CD24 positive, so analysis for additional B cell markers, such as CD79a, may be helpful [18,19].

## 3. T-Lymphoblastic Leukemia/Lymphoma

T-lymphoblastic leukemia/lymphoma is characteristically positive for T cell markers, including cytoplasmic CD3 (which is lineage-defining and specific), CD5, CD7, and CD2, which can all be assessed by flow cytometric analysis [3,20,21,22,23]. Neoplastic cells typically exhibit an immature T cell immunophenotype, which includes the expression of markers of T cell immaturity, such as TdT, and low-level expression of CD45, although the level of CD45 expression is typically greater than that seen in B-lymphoblastic leukemia/lymphoma [3,20,21,22,23]. In contrast to B-lymphoblastic leukemia/lymphoma, T-lymphoblastic leukemia/lymphoma is typically negative for HLA-DR. Cases of T-lymphoblastic leukemia/lymphoma may express markers reflective of the various stages of T cell maturation, including the pro-T, pre-T, cortical T, and medullary T cell stages. The earliest pro-T stage typically is positive for T cell markers CD7 and cytoplasmic CD3, but not later markers of T cell differentiation such as CD2, CD5, CD1a, CD4, CD8, and surface CD3. The pre-T stage typically is positive for T cell markers CD7 and cytoplasmic CD3, CD2, and CD5, but not later markers of T cell differentiation such as CD1a, CD4, CD8, and surface CD3. The cortical T cell stage expresses early T cell markers as well as CD1a, CD4, and CD8 (double positivity), but not surface CD3. The medullary stage expresses all of the above T cell markers, except that it exhibits single positivity for either CD4 or CD8 [3,20,21,22,23]. Flow cytometric findings from a typical case of T-lymphoblastic leukemia/lymphoma, corresponding to the cortical T cell stage, from a 47-year-old woman who presented with a large mediastinal mass, are shown in Figure 5. The neoplastic cells are dimly positive for CD45, shown in red, in contrast to mature lymphocytes, which exhibit increased staining for CD45 and are shown in green. The neoplastic cells were negative for HLA-DR and surface CD3, and positive for CD5, CD7, cytoplasmic CD3, CD4, CD8, CD2 CD1a, and TdT. The neoplastic cells were negative for myeloid and monocytic markers (CD11b, CD13, CD14, CD33, and CD117), and B cells markers (CD19, CD10, and CD20).

NK lymphoblastic leukemia/lymphoma was previously included in the World Health Organization classification of hematolymphoid tumors as a provisional entity, but is not included in the fifth edition. It is rare, and consists of progenitor cells that express CD56 and markers of immaturity, such as CD34 and TdT, but lack lineage-specific markers of T cells, B cells, myeloid cells, or plasmacytoid dendritic cells. In one study of six cases of NK lymphoblastic leukemia/lymphoma, patients ranged from 4–72 years in age (median age 13 years) with 30–95% blasts (median 88% blasts) present in typically hypercellular bone marrow biopsies. The neoplastic cells were positive for CD56, CD34, and CD7 in all cases, 50% were positive for TdT, and all were negative for CD19, CD123, and CD4 [24].

CD10, an immunophenotypic marker of B-lymphoblastic leukemia/lymphoma and follicular lymphoma, has been reported to be expressed in 18–63% of cases of T-lymphoblastic leukemia/lymphoma, as well as in other T cell neoplasms [6]. Another B cell marker, CD79a, may also be expressed in a minority of cases (10%) of T-lymphoblastic leukemia/lymphoma [6,20]. Myeloid antigens, including CD13, CD33, and CD15, may be expressed in a minority of cases of T-lymphoblastic leukemia/lymphoma [23]. In one study of 180 pediatric cases of T-lymphoblastic leukemia/lymphoma, myeloid antigens were expressed in 16.3% of cases; most commonly CD33, in 13% of cases, followed by CD13 in 7.1% of cases, with smaller percentages of cases expressing CD14 and CD15. Multiple myeloid antigens were expressed in 2.6% of cases [23]. It is important to note that if an acute leukemia exhibits a T cell immunophenotype (including expression of the T cell lineage-defining antigen CD3) and exhibits high-level expression of the myeloid lineage-defining marker myeloperoxidase (MPO), monocytic lineage-defining markers, or sufficient B lineage-defining markers, then a diagnosis of mixed-phenotype acute leukemia (MPAL), discussed elsewhere in this publication, is favored [8].

An additional subtype of T-lymphoblastic leukemia/lymphoma, early T-precursor lymphoblastic leukemia/lymphoma (ETP-ALL), corresponds to an early or minimal state of T cell differentiation, and expresses the T cell lineage-defining marker CD3 as well as myeloid and/or stem cell markers, but not the myeloid lineage-defining marker myeloperoxidase (<3% positive blasts) [25,26,27]. It accounts for approximately 15% of T-lymphoblastic leukemia/lymphoma cases. Neoplastic cells are typically positive for cytoplasmic CD3, negative for CD1a and CD8 (<5% positive blasts), negative or dimly positive for CD5 (<75% positive blasts), and positive for myeloid antigens, including CD11b, CD13, CD33, CD65, CD117, and stem cells antigens CD34 or HLA-DR (≥25% positive blasts). Cases with CD5 expression in ≥75% of blasts are defined as near-ETP-ALL [26]. A case of ETP-ALL, from a 3 y.o. girl with a white blood cell count of 270,000/μL with 95% circulating blasts, is shown in Figure 6. The neoplastic cells were positive for CD45, CD7 (subset in blue), CD34, cytoplasmic CD3, CD33, CD11b, and CD117, and negative for surface CD3, CD5, CD4, CD8, and CD1a. This case fulfills the criteria for ETP-ALL based on the positivity for cytoplasmic CD3, absence of staining for CD1a, CD5, and CD8, and positivity for CD11b, CD33, CD34, and CD117.

T-lymphoblastic leukemia/lymphoma, particularly cases presenting with a mediastinal mass, must be distinguished from the non-neoplastic T cells present in normal or hyperplastic thymus and thymoma [28,29]. Non-neoplastic thymic T cells exhibit the full range of developing T cells, and can be recognized by the presence of cells corresponding to all stages of thymic T cell development. Typically, thymic T cells exhibit variable staining for CD3 and CD5, and a large population of CD4/CD8 double-positive cells is present, with a minority of cells showing staining for CD4 or CD8 alone—with a characteristic smeared or double-tailed comet appearance, as well as a small population of CD4/CD8 double-negative cells. In contrast, in T-lymphoblastic leukemia/lymphoma, the neoplastic cells may be negative for surface CD3 staining or positive, but variable CD3 is usually not observed, and the expression of pan T cell antigens, including CD2, CD5, and CD7, may be absent, but not variable. In addition, the neoplastic cells in T-lymphoblastic leukemia/lymphoma do not exhibit variable CD4 and CD8 staining. Instead, neoplastic cells may be dual negative, dual positive, or positive for CD4 or CD8, typically with a tight clustering of the cells [28,29].

Recently, a newly described antibody to the T cell receptor beta chain constant region 1 (TRBC1) has been used to distinguish normal T cells from neoplastic T cells from a range of T cell neoplasms, including peripheral T cell lymphomas (including angioimmunoblastic T cell lymphoma, adult T cell lymphoma/leukemia, and peripheral T cell lymphoma, NOS), T cell large granular lymphocytic leukemia, T cell prolymphocytic leukemia, and Sezary syndrome, by flow cytometric analysis of peripheral blood, bone marrow, and lymphoid tissues [30]. TRBC1 immunophenotyping correlates with molecular testing for T cell clonality. Testing for surface TRBC1 expression requires the presence of surface CD3, which may not be seen in immature T cells. Recently, it was found that cases of T-lymphoblastic leukemia/lymphoma, which were completely or partially negative for surface CD3 and predominantly negative for surface TRBC1, were positive for cytoplasmic CD3 expression, and exhibited a monotypic positive or negative staining pattern for cytoplasmic TRBC1 [31]. Cytoplasmic TRBC1 immunophenotyping in cases of T-lymphoblastic leukemia/lymphoma correlated with molecular testing for T cell clonality. Normal thymocytes and T cells from cases of indolent T lymphoblastic proliferation were all positive for surface CD3 and exhibited a polytypic staining pattern for surface TRBC1 [31]. These findings suggest that flow cytometric analysis of cytoplasmic TRBC1 expression is useful for the assessment of T cell clonality in specimens such as mediastinal biopsies that contain immature T cells as well as other specimens under evaluation for possible involvement by T-lymphoblastic leukemia/lymphoma. Very recently, a new antibody to the T cell receptor beta chain constant region 2 (TRBC2) has been described and reported to improve the flow cytometric detection of neoplastic T cells when used in conjunction with an antibody for TRBC1 in an 11-color, single-tube T cell panel [32].

Cases of T-lymphoblastic leukemia/lymphoma exhibiting T cell markers of late-stage differentiation (medullary stage) may have a phenotype that partially overlaps with that of mature T cell neoplasms, including the expression of surface CD3 [3,20,21,22]. However, in T-lymphoblastic leukemia/lymphoma, the neoplastic cells typically express one or more markers of immaturity, such as TdT or CD34, and typically exhibit strong staining for CD7, in contrast to the absence of TdT and CD34 expression and decreased or absent CD7 expression observed in many mature T cell neoplasms [3,20,21,22,33]. Rarely, T-lymphoblastic leukemia/lymphoma with an entirely mature immunophenotype, lacking expression of TdT, CD34, CD1a, and CD99, has been reported in children, and has been associated with poor prognosis, as have cases of T-lymphoblastic leukemia/lymphoma that are TdT negative [34]. In one series, cases of T-lymphoblastic leukemia/lymphoma with an entirely mature immunophenotype were identified, in part based on the presence of neoplastic cells with a blastoid appearance, which were present in peripheral blood, bone marrow, and/or body fluids, such as pleural fluid and cerebrospinal fluid [34].

The typical flow cytometric approach for the assessment of T-lymphoblastic leukemia/lymphoma MRD includes the identification of neoplastic cells with decreased expression of CD45, expression of CD34, absence of expression of CD4 and CD8 or dual expression of these markers, overexpression of CD7, or aberrant expression of one or more myeloid or B cell markers identified at the time of diagnosis. Additional markers that may be helpful for MRD assessment include CD99, which may be overexpressed in T-lymphoblastic leukemia/lymphoma compared with normal T cells, and CD48, which exhibits reduced expression in T-lymphoblastic leukemia/lymphoma compared with normal T cells [35,36,37].

## 4. Differential Diagnosis

The differential diagnosis of B- and T-lymphoblastic leukemia/lymphoma includes acute leukemias of ambiguous lineage (ALALs), which include undifferentiated acute leukemia and mixed-phenotype acute leukemia (MPAL) [8]. In addition, acute myeloid leukemia (AML) may express B cell or T cell antigens and may be confused with B- or T-lymphoblastic leukemia/lymphoma [38]. ALAL cases are uncommon, accounting for less than 5% of acute leukemias, and, by definition, in the fifth edition of the World Health Organization classification of hematolymphoid tumors, include ≥20% abnormal progenitor cells [8].

Undifferentiated acute leukemia lacks the expression of lineage-specific markers, including cytoplasmic CD3, myeloperoxidase, multiple B cell markers including CD19, or markers of other lineages, such as NK cells, megakaryocytes, basophils, or plasmacytoid dendritic cells (Table 3) [39]. Blasts may express no more than one lineage-associated marker of any specific lineage. For example, they may be positive for one myeloid associated marker, such as CD13, CD33, or CD117, but are negative for the myeloid lineage-defining marker myeloperoxidase. Cases of acute leukemia with ≥2 myeloid-associated markers, such as CD13, CD33, and CD117, are considered acute myeloid leukemia with minimal differentiation. Undifferentiated acute leukemia may express a single B lineage-associated marker or T lineage-associated marker other than CD3 [39].

MPAL cases may express markers of B cell differentiation, T cell differentiation, and/or myeloid differentiation, and are best assessed by flow cytometric analysis (Table 3) [8]. The B cell lineage-defining markers are CD19 and CD10, CD22, or CD79a, if CD19 expression is strong (greater than 50% of that of normal B cell progenitor cells by flow cytometry), or CD19 and two of the other B cell markers, if CD19 expression is weak. If the second lineage under consideration is T, then CD79a cannot be employed as a supporting B cell marker. The T cell lineage-defining marker is cytoplasmic or surface CD3. The myeloid lineage-defining marker is myeloperoxidase, with intensity greater than 50% of the level of mature neutrophils, or two or more markers of monocytic differentiation, which include non-specific esterase, CD11c, CD14, CD64, and lysozyme [8]. MPAL cases may be biphenotypic, if they have a single population of abnormal progenitor cells expressing two or more lineages, or bilineal (or trilineal), if they have two (or three) populations of abnormal progenitor cells, each of a different lineage. The total percentage of abnormal progenitor cells must be ≥20% [8]. In one study of 100 cases of MPAL, 58% were B/myeloid, 36% were T/Myeloid, 4% were B/T, and 2% were B/T/myeloid. Another subtype, T/megakaryocytic, which expresses megakaryocytic markers (CD41, CD42, and/or CD61) along with CD3, is extremely rare [40,41].

Acute leukemia of ambiguous lineage not otherwise specified (NOS), is a rare subtype that does not express lineage-specific markers, and instead expresses combinations of markers that do not meet the criteria for acute undifferentiated acute leukemia or MPAL. An example would be an acute leukemia that expresses T lineage markers, but not CD3, myeloid lineage markers, but not myeloperoxidase, and/or B lineage marker CD19, but not other B lineage markers [42].

As noted earlier, cases of acute myeloid leukemia may express B cell or T cell antigens. In one study, 22% of acute myeloid leukemia cases expressed one or more lymphoid antigens [38]. The B cell-associated markers CD19 and CD20 were expressed on two cases (13%) and one (6%) case, respectively. Cases that were positive for CD19 did not express other B cell antigens. The T cell-associated markers CD7, CD2, and CD5 were expressed on seven (44%), six (38%), and four (25%) cases, respectively [38]. By definition, none of the cases exhibited expression of lineage-defining markers such as CD3 or multiple B cell markers including CD19, which, if present, would fulfill the diagnostic criteria for mixed-phenotype acute leukemia. As noted earlier, acute myeloid leukemia with t(8;21)(q22;q22.1) typically expresses B cell antigens, including PAX5, CD19, and CD79a. In one study of cases of acute myeloid leukemia with t(8;21)(q22;q22.1), 75% were positive for B cell marker PAX5 by immunohistochemical staining, 81% were positive for CD19 by flow cytometric analysis, with partially positive or positive expression by 11–96% of blasts (mean 42%), and 24% were positive for CD79a, with partially positive or positive expression by 11–45% of blasts (mean 25%) [5]. None of the acute myeloid leukemia with t(8;21)(q22;q22.1) cases were positive for CD20 or CD22 [5]. At least some cases exhibited staining for CD19 and CD79a, including at least one case with strong CD19 expression. However, these cases do not qualify as mixed-phenotype acute leukemia because acute leukemias that can be assigned to a specific well-characterized category, such as acute myeloid leukemia with t(8;21)(q22;q22.1), are excluded from consideration as mixed-phenotype acute leukemia.

## 5. Future Directions

Increasingly multiparametric immunophenotyping of lymphoblastic leukemias/lymphomas should improve the immunophenotypic characterization of neoplastic cells at the time of diagnosis as well as the post-therapeutic assessment of minimal or measurable residual disease (MRD). These approaches include new technologies such as spectral flow cytometry and mass cytometry, which may be combined with computer algorithms for the assessment of increasingly complex multiparametric flow cytometry data [43,44]. Computer algorithms that have been employed for the assessment of lymphoblastic leukemia/lymphoma immunophenotypic findings at the time of diagnosis and for MRD assessment include FlowSOM and viSNE [45,46]. FlowSOM enables unsupervised hierarchical clustering of cell subsets with a shared immunophenotype [45]. viSNE enables two-dimensional visual mapping of single cell data in multiparametric space to identify specific subpopulations, including rare populations [46].

## 6. Conclusions

This review has summarized the typical flow cytometric immunophenotypic findings in B- and T-lymphoblastic leukemia/lymphoma. Additional flow cytometric findings of importance that may be observed in B-lymphoblastic leukemia/lymphoma include the expression of one or more antigens of myeloid differentiation and the absence of expression of CD45 by neoplastic cells, which can interfere with the gating of the cells of interest. A number of subtypes of B-lymphoblastic leukemia/lymphoma with specific genetic abnormalities, which may correlate with specific immunophenotypic findings as well as prognosis, have been discussed. An approach to distinguish normal B cell precursors (hematogones) from residual B-lymphoblastic leukemia/lymphoma following chemotherapy is described, as well as new approaches to identify residual B-lymphoblastic leukemia/lymphoma following immunotherapy with various agents.

The typical flow cytometric immunophenotypic findings in T-lymphoblastic leukemia/lymphoma, NK lymphoblastic leukemia/lymphoma, and early T-precursor lymphoblastic leukemia/lymphoma (ETP-ALL) have been summarized. Additional flow cytometric findings of importance that may be observed in T-lymphoblastic leukemia/lymphoma include the expression of one or more antigens of B cell or myeloid differentiation. An approach to distinguish T-lymphoblastic leukemia/lymphoma from the non-neoplastic T cells present in normal or hyperplastic thymus and thymoma has been described.

Finally, the flow cytometric findings in a number of neoplasms that may be difficult to distinguish from B- and T-lymphoblastic leukemia/lymphoma are discussed. These include undifferentiated acute leukemia, mixed-phenotype acute leukemia (MPAL), acute leukemia of ambiguous lineage not otherwise specified (NOS), and acute myeloid leukemia with expression of B cell or T cell antigens.

## Figures and Tables

**Figure 1 cancers-17-01111-f001:**
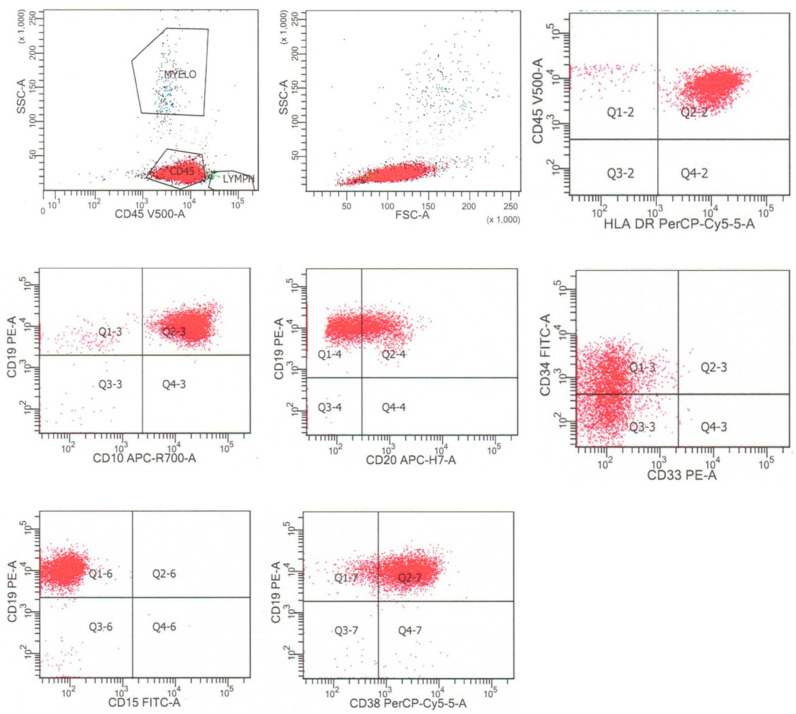
Flow cytometry of a typical case of B-lymphoblastic leukemia/lymphoma, that is CD45(dim), HLA-DR+, CD19+, CD10+, CD20dim+, CD34dim+, and CD38+, and negative for CD15 and other myeloid antigens. The neoplastic cells are dimly positive for CD45 and are colored red.

**Figure 2 cancers-17-01111-f002:**
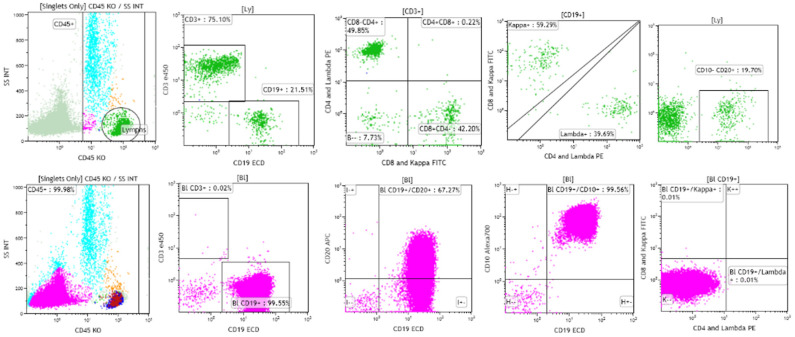
A case of CD45-negative B-lymphoblastic leukemia/lymphoma from a 4-year-old girl with pancytopenia and increased bone marrow blasts. Gating on CD45-positive cells reveals the presence of polyclonal B cells, and CD4 and CD8-positive T cells (**top** row). Gating on CD45-negative cells reveals the presence of CD19 and CD10 positive blasts that are dimly positive for CD20 and negative for surface immunoglobin kappa and lambda light chains (**bottom** row). Case courtesy of Mr. Mike Keeney, Ontario Health Sciences Center, London, Ontario.

**Figure 3 cancers-17-01111-f003:**
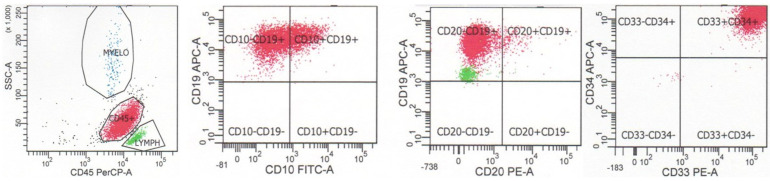
Flow cytometry of a case of B-lymphoblastic leukemia/lymphoma with BCR::ABL1 fusion. The neoplastic cells are dimly positive for CD45, positive for CD34, positive for B cell markers CD19 and CD10, show dim-to-negative staining for CD20, and coexpress myeloid markers CD33 and CD13.

**Figure 4 cancers-17-01111-f004:**
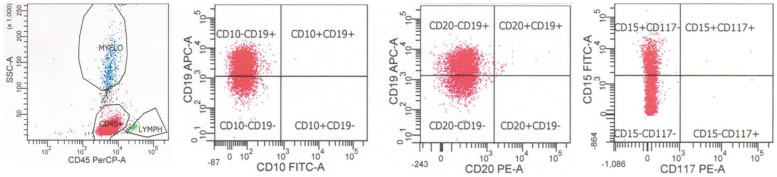
Flow cytometry of a case of B-lymphoblastic leukemia/lymphoma with *KMT2A* rearrangement. The neoplastic cells are dimly positive for CD45, positive for B cell marker CD19, negative for B cell markers CD10 and CD20, and coexpresses myeloid marker CD15 (dim).

**Figure 5 cancers-17-01111-f005:**
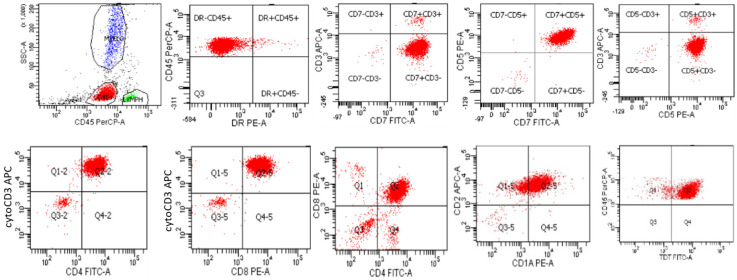
Flow cytometry of a typical case of T-lymphoblastic leukemia/lymphoma, corresponding to the cortical T cell stage, that is CD45 (dim), negative for HLA-DR and surface CD3, and positive for CD5, CD7, cytoplasmic CD3, CD4, CD8, CD2 CD1a, and TdT.

**Figure 6 cancers-17-01111-f006:**
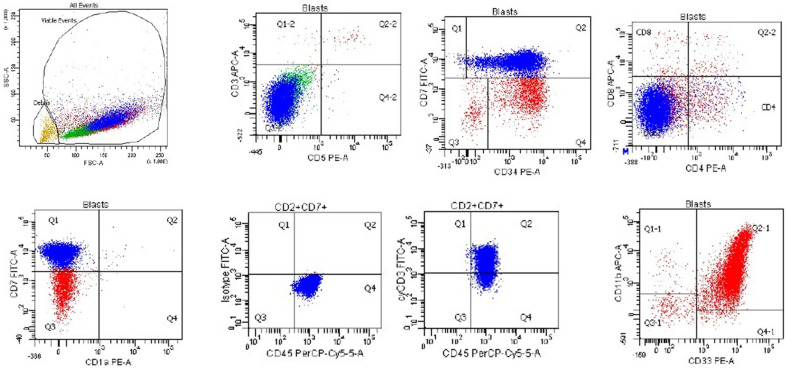
Flow cytometry of a case of early T-precursor lymphoblastic leukemia/lymphoma from a 3-year-old girl with leukocytosis (WBC = 270,000/μL with 95% blasts) that is positive for CD45, CD7 (subset in blue), CD34, cytoplasmic CD3, CD33, CD11b, and CD117, and negative for surface CD3, CD5, CD4, CD8, and CD1a. Case courtesy of Dr. Olga Weinberg, Boston Children’s Hospital, Boston, Massachusetts.

**Table 1 cancers-17-01111-t001:** Specific immunophenotypic findings in B-lymphoblastic leukemia/lymphoma with recurrent genetic abnormalities [4].

B-lymphoblastic leukemia/lymphoma with BCR::ABL1 fusion
Coexpression of myeloid markers CD33 and CD13; decreased expression of CD9 and CD81.
B-lymphoblastic leukemia/lymphoma with BCR::ABL1-like features
Overexpression of CRLF2 protein.
B-lymphoblastic leukemia/lymphoma with KMT2A rearrangement
CD19-positive, CD10-negative, CD24-negative, and TdT-negative immunophenotype, as well as positivity for myeloid markers CD15 and CD65.
B-lymphoblastic leukemia/lymphoma with ETV6::RUNX1 fusion
Often negative for CD9, CD20, and CD66c, and positive for myeloid markers CD13 and CD33.
B-lymphoblastic leukemia/lymphoma with TCF3::PBX1 fusion
Typically strongly positive for CD9, dim-to-negative for CD34, and dim-to-negative for CD20.
B-lymphoblastic leukemia/lymphoma with TCF3::PBX1 fusion
Overexpression of CD58 and CD81.
B-lymphoblastic leukemia/lymphoma with IGH::IL3 fusion
Expression of myeloid markers CD13 and/or CD33.
B-lymphoblastic leukemia/lymphoma with DUX4 re-arrangement
Typically positive for CD371 and CD2; cases may exhibit expressions of monocytic marker CD14, gains of CD45 and CD33, and losses of B cell antigens after induction chemotherapy or at the time of diagnosis.

**Table 2 cancers-17-01111-t002:** Immunophenotypic differences between hematogones and B-lymphoblastic leukemia/lymphoma [4].

Hematogone	B-Lymphoblastic Leukemia/Lymphoma.
Intermediate/variable CD45	Absent, intermediate, or bright CD45.
Moderate expression of CD10	Brighter expression of CD10 or absence of CD10 expression.
Bright CD38	May be weak or absent CD38.
Variable CD20	Positive, variable, or absent CD20.
Subset CD34 positive	CD34 positive or negative.
Negative for myeloid antigens	May express one or more myeloid antigens.

**Table 3 cancers-17-01111-t003:** Immunophenotypic features of acute leukemias of ambiguous lineage.

	Myeloid Markers ^1^	B Cell Markers ^2^	T Cell Markers ^3^
Acute undifferentiated leukemia ^4^	−	−	−
MPAL, B/Myeloid	+	+	−
MPAL, T/Myeloid	+	−	+
MPAL, B/T	−	+	+
MPAL, B/T/Myeloid	+	+	+

^1^ Myeloperoxidase (MPO) or two or more markers of monocytic differentiation (non-specific esterase, CD11c, CD14, CD64, and lysozyme). ^2^ CD19 and CD10, CD22 or CD79a if CD19 expression is strong, or CD19 and two of the other B cell markers, if CD19 expression is weak. ^3^ Cytoplasmic or surface CD3. ^4^ Negative for CD3, MPO, multiple B cell markers, including CD19, and markers of other lineages.

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
