# Peer review of "The Flow Cytometric Evaluation of B- and T-Lymphoblastic Leukemia/Lymphoma"

_cancers, 2025, doi:10.3390/cancers17071111_

Round 1
Reviewer 1 Report
Comments and Suggestions for Authors
This is a comprehensive review of the current status of evaluation of B and T lymphoblastic leukemia/lymphoma by flow cytometry. It is clear and easy to follow with good examples been illustrated and described the challenges with utility of flow cytometry in assessing treatment response due to the increasing availability of newer treatment in particular immunotherapies which required modification of antibody panel and gating strategies in assessing MRD.
Suggest inclusion of a few tables, if possible, to highlight some of the key messages, such as the differences between hematogones and residual B-lymphoblastic leukemia/lymphoma, and the subtypes categorized by genetic abnormalities for B-ALL, T-lymphoblastic leukemia/lymphoma. This will reinforce the key points of the review and provide readers with easy to refer to tables for their application in practice.
The section on differential diagnosis is well summarized, as this is a typical diagnostic challenge encountered, once again, a table to help summarize the key differences can add clarity to this well written review.
Author Response
This is a comprehensive review of the current status of evaluation of B and T lymphoblastic leukemia/lymphoma by flow cytometry. It is clear and easy to follow with good examples been illustrated and described the challenges with utility of flow cytometry in assessing treatment response due to the increasing availability of newer treatment in particular immunotherapies which required modification of antibody panel and gating strategies in assessing MRD.
Suggest inclusion of a few tables, if possible, to highlight some of the key messages, such as the differences between hematogones and residual B-lymphoblastic leukemia/lymphoma, and the subtypes categorized by genetic abnormalities for B-ALL, T-lymphoblastic leukemia/lymphoma. This will reinforce the key points of the review and provide readers with easy to refer to tables for their application in practice.
Requested tables have been added.
The section on differential diagnosis is well summarized, as this is a typical diagnostic challenge encountered, once again, a table to help summarize the key differences can add clarity to this well written review.
Requested table has been added.
Reviewer 2 Report
Comments and Suggestions for Authors
The author reviews the flow cytometric analysis of B- and T-acute lymphoblastic leukemia.
It is a well-known area with many reviews in the literature.
Comments:
-There is a reference list, but it is unclear which citation belongs to which part, as there are no reference numbers or authors' names in the text!
- The quality of the figures is poor. The gating is not accurate in Fig 1 and 4. The figures were made by different software and they are not in the same style.
- The legend of Figure 6. repeats the main text. It doesn't show the CD117 expression of EATL blasts, which is an important, differential diagnostic marker of them.
- The of Figure 5 is not classified, however, it is easily classifiable.
- B-ALL cases described as mainly CD45 positive and later T-ALL cases express low CD45. It is a contradiction because the CD45 expression of T-ALL blasts is higher in most cases as described on page 3.
- CD99 and CD48 are useful markers of T-ALL blasts, but they are not mentioned.
- The author mentioned TRCB1 but not TRCB2. Why? This marker is also available on the market now.
- Which marker can be used for MRD assessment? Which is reliable and which is not?
- Some tables should have been useful about the groups.
Author Response
The author reviews the flow cytometric analysis of B- and T-acute lymphoblastic leukemia.
It is a well-known area with many reviews in the literature.
Comments:
-There is a reference list, but it is unclear which citation belongs to which part, as there are no reference numbers or authors' names in the text!
Superscripts with citations have been added to the text.
- The quality of the figures is poor. The gating is not accurate in Fig 1 and 4. The figures were made by different software and they are not in the same style.
The gating in figures 1 and 4 is on the lymphoblasts that are dimly positive for CD45.
- The legend of Figure 6. repeats the main text. It doesn't show the CD117 expression of EATL blasts, which is an important, differential diagnostic marker of them.
An image of CD117 expression is not available. CD117 positivity is mentioned in the text. Positivity for CD11b and CD33 is shown in the figure.
- The of Figure 5 is not classified, however, it is easily classifiable.
A subtype has been added to the text.
- B-ALL cases described as mainly CD45 positive and later T-ALL cases express low CD45. It is a contradiction because the CD45 expression of T-ALL blasts is higher in most cases as described on page 3.
B-LL exhibits low level expression of CD45 (line 47). Some B-LL cases may be negative for CD45 (lines 90-91). T-LL exhibits low-level expression of CD45, although the level of CD45 expression is typically greater than that seen in B-lymphoblastic leukemia/lymphoma. (lines 208-209).
- CD99 and CD48 are useful markers of T-ALL blasts, but they are not mentioned.
CD99 and CD48 are now mentioned in the text.
- The author mentioned TRCB1 but not TRCB2. Why? This marker is also available on the market now.
A reference to TRBC2 is now included.
- Which marker can be used for MRD assessment? Which is reliable and which is not?
A detailed discussion of MRD assessment is beyond the scope of this review.
- Some tables should have been useful about the groups.
Tables have been added.
Reviewer 3 Report
Comments and Suggestions for Authors
The review article by D.M. Dorfman discusses advantages and problems of the flow cytometric diagnosis of B- and T-lymphoblastic leukemia/ lymphoma.
GENERAL COMMENTS
The present paper is mostly set on a clinical and practical basis, with only a limited discussion on the (many) technical aspects of the flow cytometric analysis. The readership will be presumably made by oncologists and practitioners, so the overall tone of the present manuscript seems well balanced, highlighting the fundamental role of the flow cytometric analysis in the quick diagnosis and differential diagnosis of acute lymphocytic malignancies.
As a review paper on a complex and ever-evolving topic, one would expect a more detailed and varied list of references, since B- and T-lymphoblastic leukemia/lymphomas are characterized by a great number of variants or aberrant presentations, and an endless flow of reports describes cases with unexpected or ambiguous phenotypes.
In a review article of this level one would also expect a discussion or comparison among the different antibody mixtures/panels devised over the recent years by qualified international consortia like EUROFLOW (Bras AE, et al. CD123 expression levels in 846 acute leukemia patients based on standardized immunophenotyping. Cytometry B Clin Cytom 2019 Mar; 96(2): 134-142), AIEOP (Conter V, et al. Early T-cell precursor acute lymphoblastic leukaemia in children treated in AIEOP centres with AIEOP-BFM protocols: a retrospective analysis. Lancet Haematol 2016 Feb;3(2):e80-6. doi: 10.1016/S2352-3026(15)00254-9) and ELN (Gökbuget N, et al. Diagnosis, prognostic factors, and assessment of ALL in adults: 2024 ELN recommendations from a European expert panel. Blood 2024 May 9; 143(19): 1891-1902).
A more in-depth distinction between pediatric and adult cases seems sometimes necessary, because in the two clinical settings different clinical presentation and different immunophenotypes can be found (See: Kowarsch F, et al. FCM marker importance for MRD assessment in T-cell acute lymphoblastic leukemia: An AIEOP-BFM-ALL-FLOW study group report. Cytometry A 2024 Jan; 105(1): 24-35. doi: 10.1002/cyto.a.24805). The laboratories involved in the diagnosis and follow-up of pediatric cases are also used to manage different antibody panels, as compared to labs dealing with adult cases.
SPECIFIC COMMENTS
The whole main text lacks reference numbers, which is quite bothering for the reader. Please amend. In the main text "...in one study..." or "...has been reported..." were mentioned several times, without the possibility to find the right paper in the reference list.
Chapter 2. B-lymphoblastic leukemia/lymphoma, line 51: At least one reference on PAX5 is warranted, since this marker is still not very popular in clinical diagnostics (for example: Gu Z et al. PAX5-driven subtypes of B-progenitor acute lymphoblastic leukemia. Nat Genet 2019 Feb; 51(2): 296-307. doi: 10.1038/s41588-018-0315-5).
The fact that in T-cell lymphoblastic leukemias some antigens may undergo changes during therapy (i.e. TdT, CD99) should be mentioned. This can impact on MRD studies.
The discussion of the case shown in Figure 2 may deserve a different wording. The take home message is that lymphoblastic leukemia cells can display the lowest density of the CD45 antigen, so no threshold nor gating windows should be set on this parameter to include the whole spectrum of CD45 positive or negative cells, as any well-trained cytometrist knows. Please reword the paragraph in the main text from line 91 to line 104 and the legend of Figure 2.
The term 'minimal residual disease' should be turned into 'measurable residual disease' wherever applicable, to comply with the currently accepted terminology.
Chapter 3, T-lymphoblastic leukemia/lymphoma, line 279 on: When discussing the role of TRBC1, it is necessary to include also TRBC2, that allows the full evaluation of T cell clonality and replacing reference n. 21 (See: Horna P, et al. Dual T-cell constant β chain (TRBC)1 and TRBC2 staining for the identification of T-cell neoplasms by flow cytometry. Blood Cancer J 2024 Feb 29;14(1):34. doi: 10.1038/s41408-024-01002-0).
Chapter 5, Future directions / concluding remarks. The existence of fully BS EN ISO/IEC 17043:2010 accredited programs for Measurable Residual Disease for ALL by Flow Cytometry (i.e: https://www.ukneqasli.co.uk/eqa-pt-programmes/flow-cytometry-programmes/) should be cited, to highlight the high clinical value, reliability and utility of MRD studies in T- and B-ALL.
Author Response
The review article by D.M. Dorfman discusses advantages and problems of the flow cytometric diagnosis of B- and T-lymphoblastic leukemia/ lymphoma.
GENERAL COMMENTS
The present paper is mostly set on a clinical and practical basis, with only a limited discussion on the (many) technical aspects of the flow cytometric analysis. The readership will be presumably made by oncologists and practitioners, so the overall tone of the present manuscript seems well balanced, highlighting the fundamental role of the flow cytometric analysis in the quick diagnosis and differential diagnosis of acute lymphocytic malignancies.
As a review paper on a complex and ever-evolving topic, one would expect a more detailed and varied list of references, since B- and T-lymphoblastic leukemia/lymphomas are characterized by a great number of variants or aberrant presentations, and an endless flow of reports describes cases with unexpected or ambiguous phenotypes.
References have been added to the text as superscripts. A number of important clinical variants are discussed, but it is beyond the scope of the review to include all cases with unexpected or ambiguous phenotypes.
In a review article of this level one would also expect a discussion or comparison among the different antibody mixtures/panels devised over the recent years by qualified international consortia like EUROFLOW (Bras AE, et al. CD123 expression levels in 846 acute leukemia patients based on standardized immunophenotyping. Cytometry B Clin Cytom 2019 Mar; 96(2): 134-142), AIEOP (Conter V, et al. Early T-cell precursor acute lymphoblastic leukaemia in children treated in AIEOP centres with AIEOP-BFM protocols: a retrospective analysis. Lancet Haematol 2016 Feb;3(2):e80-6. doi: 10.1016/S2352-3026(15)00254-9) and ELN (Gökbuget N, et al. Diagnosis, prognostic factors, and assessment of ALL in adults: 2024 ELN recommendations from a European expert panel. Blood 2024 May 9; 143(19): 1891-1902).
A technical discussion of antibody mixtures/panels is beyond the scope of this review.
A more in-depth distinction between pediatric and adult cases seems sometimes necessary, because in the two clinical settings different clinical presentation and different immunophenotypes can be found (See: Kowarsch F, et al. FCM marker importance for MRD assessment in T-cell acute lymphoblastic leukemia: An AIEOP-BFM-ALL-FLOW study group report. Cytometry A 2024 Jan; 105(1): 24-35. doi: 10.1002/cyto.a.24805). The laboratories involved in the diagnosis and follow-up of pediatric cases are also used to manage different antibody panels, as compared to labs dealing with adult cases.
A detailed discussion of MRD and MRD differences between pediatric and adult cases is beyond the scope of this review.
SPECIFIC COMMENTS
The whole main text lacks reference numbers, which is quite bothering for the reader. Please amend. In the main text "...in one study..." or "...has been reported..." were mentioned several times, without the possibility to find the right paper in the reference list.
Reference numbers have been added to the text as superscripts.
Chapter 2. B-lymphoblastic leukemia/lymphoma, line 51: At least one reference on PAX5 is warranted, since this marker is still not very popular in clinical diagnostics (for example: Gu Z et al. PAX5-driven subtypes of B-progenitor acute lymphoblastic leukemia. Nat Genet 2019 Feb; 51(2): 296-307. doi: 10.1038/s41588-018-0315-5).
The reference is to genetic alterations in the PAX5 gene and transcriptome sequencing to classify B-LL, which is beyond the scope of this review.
The fact that in T-cell lymphoblastic leukemias some antigens may undergo changes during therapy (i.e. TdT, CD99) should be mentioned. This can impact on MRD studies.
A discussion of phenotypic alterations following therapy and the consequences for MRD assessment is beyond the scope of this review.
The discussion of the case shown in Figure 2 may deserve a different wording. The take home message is that lymphoblastic leukemia cells can display the lowest density of the CD45 antigen, so no threshold nor gating windows should be set on this parameter to include the whole spectrum of CD45 positive or negative cells, as any well-trained cytometrist knows. Please reword the paragraph in the main text from line 91 to line 104 and the legend of Figure 2.
The example is a case of CD45-negative B-LL, which in typical practice may be missed using standard gating, as this case from an academic medical center with well-trained cytometrists illustrates. The suggestion that no threshold nor gating windows should be set for CD45 is not standard clinical practice.
The term 'minimal residual disease' should be turned into 'measurable residual disease' wherever applicable, to comply with the currently accepted terminology.
This change has been made in the text.
Chapter 3, T-lymphoblastic leukemia/lymphoma, line 279 on: When discussing the role of TRBC1, it is necessary to include also TRBC2, that allows the full evaluation of T cell clonality and replacing reference n. 21 (See: Horna P, et al. Dual T-cell constant β chain (TRBC)1 and TRBC2 staining for the identification of T-cell neoplasms by flow cytometry. Blood Cancer J 2024 Feb 29;14(1):34. doi: 10.1038/s41408-024-01002-0).
The reference to TRBC2 is now included in the text.
Chapter 5, Future directions / concluding remarks. The existence of fully BS EN ISO/IEC 17043:2010 accredited programs for Measurable Residual Disease for ALL by Flow Cytometry (i.e: https://www.ukneqasli.co.uk/eqa-pt-programmes/flow-cytometry-programmes/) should be cited, to highlight the high clinical value, reliability and utility of MRD studies in T- and B-ALL.
A detailed discussion of MRD is beyond the scope of this review.
Round 2
Reviewer 2 Report
Comments and Suggestions for Authors
The quality of Figures haven't been corrected. The gating is not appropriate in Fig. 1 and Fig. 4. (granulocyte gate, lymphocyte gate)
Gatings were performed by different software at different figures. It should be uniformed. The quadrant labelling is not neccesary on the dot-plots.
TRBC2 is available on the marker by different vendors! So line 324 is not true!
Table 3 is missleading. It says there are no myeloid or other markers on AUL, but it is not true based on the text. It would be better if the notes (markers) presented in the table and not just notes.
Author Response
The quality of Figures haven't been corrected. The gating is not appropriate in Fig. 1 and Fig. 4. (granulocyte gate, lymphocyte gate)
The figures are of classic cases that I have in my files that illustrate the flow cytometric findings in typical B lymphoblastic leukemia/lymphoma (Figure 1) and B-lymphoblastic leukemia/lymphoma with KMT2A rearrangement (Figure 4). In each case, the cells of interest, the blasts, are correctly gated and colored red. The other cells shown (granulocytes and mature lymphocytes are not relevant to the flow cytometric findings shown for the blasts, and the flow cytometry findings for those cells are not shown in the figures. I am unable to edit the figures.
Gatings were performed by different software at different figures. It should be uniformed. The quadrant labelling is not neccesary on the dot-plots.
The figures are of classic cases that I have in my files that illustrate the typical flow cytometric findings in various types of B and T lymphoblastic leukemias/lymphomas. The cases are from different laboratories that employ different flow cytometry analysis software and display the data differently. There is no reason to require uniformity of software or quadrant labeling for this review, and in fact, showing cases from different institutions that employ different analysis software and visual displays is useful for readers to see. I am unable to edit the figures.
TRBC2 is available on the marker by different vendors! So line 324 is not true!
I have removed that statement that TRBC2 antibody is not yet commercially available.
Table 3 is missleading. It says there are no myeloid or other markers on AUL, but it is not true based on the text. It would be better if the notes (markers) presented in the table and not just notes.
Table 3 does not state that there are no myeloid markers on AUL, it indicates that AUL does not express MPO or two or more markers of monocytic differentiation.